# Fast Adiabatic Control of an Optomechanical Cavity

**DOI:** 10.3390/e25010018

**Published:** 2022-12-22

**Authors:** Nicolás F. Del Grosso, Fernando C. Lombardo, Francisco D. Mazzitelli, Paula I. Villar

**Affiliations:** 1Departamento de Física Juan José Giambiagi, FCEyN UBA and IFIBA CONICET-UBA, Facultad de Ciencias Exactas y Naturales, Ciudad Universitaria, Pabellón I, Buenos Aires 1428, Argentina; 2Centro Atómico Bariloche and Instituto Balseiro, Comisión Nacional de Energía Atómica, Bariloche 8400, Argentina

**Keywords:** quantum control, shortcuts to adiabaticity, control theory, optomechanical cavity, quantum technology

## Abstract

The development of quantum technologies present important challenges such as the need for fast and precise protocols for implementing quantum operations. Shortcuts to adiabaticity (STAs) are a powerful tool for achieving these goals, as they enable us to perform an exactly adiabatic evolution in finite time. In this paper, we present a shortcut to adiabaticity for the control of an optomechanical cavity with two moving mirrors. Given reference trajectories for the mirrors, we find analytical expressions that give us effective trajectories which implement an STA for the quantum field inside the cavity. We then solve these equations numerically for different reference protocols, such as expansions, contractions and rigid motions, thus confirming the successful implementation of the STA and finding some general features of these effective trajectories.

## 1. Introduction

Quantum technologies promise to revolutionize the way we communicate and process information by giving us the ability to experimentally manipulate quantum states of light and matter at the single-particle level [1,2,3]. To this end, it is necessary to isolate these systems from the interaction with their surroundings in such a way that it might be possible, for example, to cool atoms close to absolute zero or to maintain the fragile quantum correlations between these systems. Likewise, this degree of control of quantum systems also enables their use for more efficient information processing or as quantum simulators of complex dynamics. In this context, it is necessary to understand different aspects such as the system dynamics of many interacting quantum systems; the possible decoherence processes that these devices may undergo, and the thermodynamics of systems on these scales. A natural question has emerged about whether it is possible to use new technologies to produce quantum machines. The novelty comes from the fact that these small systems can exhibit quantum properties that could potentially be exploited to get an advantage over classical machines or present new obstacles to their operation. These questions constitute the backbone of a new area of physics that has come to be called quantum thermodynamics, a fruitful crucible of research fields where the foundations of physics, information science and statistical mechanics merge.

In most cases, a finite-time operation causes the emergence of coherence in the state of the system that results in an efficiency loss [4,5,6]. However, in many cases, it is possible to implement protocols named shortcuts to adiabaticity (STAs), that evolve the initial state into the final state that would have been obtained with an adiabatic evolution, but in a finite time [7,8,9,10]. STAs are powerful quantum control methods, allowing quick evolution into target states of otherwise slow adiabatic dynamics. Such methods have widespread applications in quantum technologies, and various shortcuts to adiabaticity protocols have been demonstrated in closed systems. These protocols typically require a full control of the quantum system and end up being extremely challenging from an experimental standpoint.

Another area where an STA might be extremely useful is relativistic quantum information (RQI). Fundamental questions have arisen on how the motion of different observers affect shared quantum information and how to distribute and process it [11,12,13,14,15,16,17]. Recent works have shown that the entanglement shared between two moving cavities is diminished as observers accelerate [12,13]. This is due to the fact that their motion causes a nonadiabatic evolution of the quantum system that generates excitations that affect the entanglement [18]. Hence, if one can find an STA that achieves a fast adiabatic evolution of the field inside a moving cavity, it would be possible to exactly preserve the entanglement solving a fundamental problem in RQI.

In previous works, STAs have been considered from a theoretical and/or an experimental point of view for different physical systems: trapped ions [19], cold atoms [20], ultracold Fermi gases [21], Bose–Einstein condensates in atom chips [22], etc. In Ref. [23], we showed how to implement shortcuts to adiabaticity for the case of a massless scalar field inside a cavity with a moving wall, in (1 + 1) dimensions. The approach was based on the known solution to the problem that exploited the conformal symmetry, and the shortcuts took place whenever the solution matched the adiabatic Wentzel–Kramers–Brillouin (WKB) solution [24], i.e., when there was no dynamical Casimir effect (DCE) [25,26,27,28,29]. We obtained a fundamental limit for the efficiency of an Otto cycle with the quantum field as a working system, which depended on the maximum velocity that the mirror could attain. We also described possible experimental realizations of the shortcuts using superconducting circuits.

In this paper, we generalize the results of [23] to the case of a quantum scalar field in a one-dimensional optomechanical cavity with two moving mirrors. We show that, given the trajectories for the left ($L_{\mathrm{ref}}\left(t\right)$) and right ($R_{\mathrm{ref}}\left(t\right)$) mirrors, we can find a shortcut to adiabaticity ruled by the effective trajectories ($L_{\mathrm{eff}}\left(t\right)$) and ($R_{\mathrm{eff}}\left(t\right)$) that, when implemented in finite time, result in the same state as if the original ones had been evolved adiabatically. This protocol has the advantage that it can be easily implemented experimentally using either an optomechanical cavity or superconducting circuits, since it does not require additional exotic potentials. Moreover, the effective trajectory can be computed from the original one quite simply, paving the way for more efficient quantum field thermal machines. Besides its intrinsic interest, this generalization may have useful applications in the area of RQI.

In the next Section we discuss that for a quantum field, STAs are not as simple as for a nonrelativistic quantum system with a finite number of degrees of freedom. Section 3 is dedicated to the study of an optomechanical cavity with two moving mirrors and, in Section 4, we show how to find STAs in these cavities. Section 5 is dedicated to the numerical analysis of the STA for different reference trajectories such as a contraction, expansion or a rigid motion of the cavity. In Section 6, we complete the work with a discussion of our results.

## 2. STA in Quantum Field Theory

When a quantum field is subjected to time-dependent external conditions, the phenomenon of particle creation seems unavoidable. However, as already mentioned, in some particular situations this phenomenon can be avoided. We shall discuss some examples in the following.

### 2.1. Electromagnetic Cavity: Single-Mode Approximation

Let us consider an electromagnetic cavity with time-dependent properties (variable length and/or time-dependent electromagnetic properties). It is usual to describe the physics inside the cavity using a single-mode approximation for the quantum electromagnetic field. The dynamics of the mode is that of a harmonic oscillator with a time-dependent frequency
(1)$${\ddot{Q}}_{k}+\omega_{k}^{2}\left(t\right)Q_{k}=0,$$
where k is the index that identifies the mode. The frequency $\omega_{k}\left(t\right)$ depends on time if, for instance, the length of the cavity d(t) is time-dependent.

Assuming that the frequency is constant for t→±∞, and that the mode is in the ground state $|0_{IN}\rangle$ for t→−∞, in the case of a nonadiabatic evolution the electromagnetic mode will be excited for t→+∞, that is $|\langle 0_{OUT}|0_{IN}\rangle |\neq 1$. The Bogoliubov transformation that connects the IN and OUT Fock spaces, when nontrivial, is an indication of particle creation and describes the presence of photons inside the cavity.

The adiabatic WKB solution for the operator associated with the mode, ${\hat{Q}}_{k}\left(t\right)$, can be written in terms of annihilation and creation operators as
(2)$${\hat{Q}}_{k}\left(t\right)=\hat{a}\frac{e^{-i\int^{t}\omega_{k\;\mathrm{ref}}\left(t^{\prime }\right)dt^{\prime }}}{\sqrt{2\omega_{k\;\mathrm{ref}}\left(t\right)}}+{\hat{a}}^{†}\frac{e^{i\int^{t}\omega_{k\;\mathrm{ref}}\left(t^{\prime }\right)dt^{\prime }}}{\sqrt{2\omega_{k\;\mathrm{ref}}\left(t\right)}}.$$
This is an approximate solution for the oscillator with a reference frequency $\omega_{\;\mathrm{ref}}\left(t\right)$, valid if it is slowly varying, but an exact solution of a system with an effective frequency [24]
(3)$$\omega_{k\;\mathrm{eff}}^{2}\left(t\right)=\omega_{k\;\mathrm{ref}}^{2}+\frac{1}{2}\left(\frac{{\ddot{\omega }}_{k\;\mathrm{ref}}}{\omega_{k\;\mathrm{ref}}}-\frac{3}{2}{\left(\frac{{\dot{\omega }}_{k\;\mathrm{ref}}}{\omega_{k\;\mathrm{ref}}}\right)}^{2}\right)\;.$$
From the effective frequency, one can read the effective time-dependent length of the cavity $d_{k\;\mathrm{eff}}\left(t\right)$ which leads to no particle creation, and therefore constitutes an STA. It is important to remark that the evolution at intermediate times is in general nonadiabatic, but the system returns to the initial state when the effective length becomes constant at t→+∞. Particles are created and subsequently absorbed.

The STA described above cannot be generalized beyond the single-mode approximation since the effective frequency and length are different for each mode, and therefore it is not possible to avoid particle creation in all modes. Moreover, for this system, an electromagnetic field inside a time-dependent cavity, the modes are coupled.

In the rest of the paper, we consider a physical system in which it is possible to find a nontrivial STA for a full quantum field. By nontrivial we mean that, although there is no particle creation at the end of the evolution, the dynamics is nonadiabatic at intermediate times, that is, there is creation and absorption of particles. Before doing this, we mention some examples of quantum fields in time-dependent backgrounds in which there is no particle creation at all, that is, the modes of the fields are oscillators with time-independent frequency.

### 2.2. Quantum Fields in Curved Space-Times

Assuming a Robertson–Walker metric
(4)$$ds^{2}=a^{2}\left(\eta \right)\left(-d\eta^{2}+dx^{2}\right)\;,$$
the modes of a free quantum scalar field satisfy [30,31]
(5)$${\ddot{\chi }}_{k}+\left(k^{2}+m^{2}a^{2}+\left(\xi -1/6\right)Ra^{2}\right)\chi_{k}=0\;,$$
where *m* is the mass of the field, *R* the scalar curvature and ξ the coupling to the curvature. We are describing the dynamical equations in terms of the conformal time η and a(η) is the scale factor. The equations for the modes correspond to those of harmonic oscillators with time-dependent frequency. As mentioned above, for each mode, one can find particular evolutions of the scale factor such that there is no particle creation. However, as the time-dependent frequency depends on the momentum *k*, it is not possible to find an STA for the full quantum field, but only for a given mode.

There are some particular situations in which the frequency of all modes is time-independent, for an arbitrary time dependence of the scale factor. This is the case when there is conformal invariance m=0 and ξ=1/6. Another possibility is to consider a massless field in a radiation-dominated universe, for which R=0 (for another example in the context of non-Abelian field theories see [32]).

The relevance of conformal invariance can be reinforced by another example. Let us consider now a massless quantum scalar field in an almost flat metric
(6)$$ds^{2}=\left(\eta_{\mu \nu }+h_{\mu \nu }\right)dx^{\mu }dx^{\nu }\;,$$
with $|h_{\mu \nu }|\ll 1$. The probability of pair creation reads [33]
(7)$$P=\frac{1}{960\pi }\int d^{4}x\left[60\left(\xi -1/6\right)R^{2}+C_{\mu \nu \rho \sigma }C^{\mu \nu \rho \sigma }\right]\;,$$
where $C_{\mu \nu \rho \sigma }$ is the Weyl tensor. Once again, for a conformal field (ξ=1/6) in a conformally flat metric $\left(C_{\mu \nu \rho \sigma }=0\right)$, the pair creation probability vanishes.

There are some subtle points in these examples. On the one hand, particle creation vanishes when one chooses the conformal vacuum as the vacuum state of the system. For Robertson–Walker metrics, this corresponds to the choice of the mode functions
(8)$$\chi_{k}=\frac{1}{\sqrt{2k}}e^{-ik\eta }\;,$$
that solve Equation (Equation 4) when m=0 and ξ=1/6. This choice is natural if the metric is asymptotically flat for η→−∞. The mean value of the energy–momentum tensor vanishes in that region. Even with this choice, it is known that conformal invariance is broken at the quantum level, producing a nonvanishing trace for the mean value of the energy–momentum tensor (that is traceless for a conformal field at the classical level). While each mode of the quantum field evolves in a trivial way, the mean value of the energy–momentum tensor may depend on time during the evolution. This dependence is, however, local in the metric and its derivatives, and therefore, the energy–momentum tensor returns to its vanishing value if the scale factor tends to a constant for η→+∞.

From the previous discussion, we see that for a quantum field, STAs are not as simple as for a quantum system with a finite number of degrees of freedom. The renormalization, which is unavoidable even for free fields in external backgrounds, is an additional ingredient that should be taken into account. On the other hand, we also see that while conformal invariance simplifies the dynamical equations for the modes, its quantum anomaly may introduce nontrivial effects. We see all these aspects at work in the optomechanical cavity with moving mirrors.

## 3. The Optomechanical Cavity

The system we are now considering is a scalar field, Φ(x,t), inside a cavity formed by two moving mirrors to the left and right whose position are given by L(t) and R(t), respectively (see Figure 1). The evolution of the field is determined by the wave equation inside the cavity
(9)$$\left(\partial_{x}^{2}-\partial_{t}^{2}\right)\Phi \left(x,t\right)=0,$$
and Dirichlet boundary conditions on each mirror
(10)Φ(L(t),t)=Φ(R(t),t)=0.

It is important to remark that we are considering units where $c=\hbar =k_{B}=1$, which we use throughout the rest of the paper. It is known that the time evolution of the field is solved by expanding the field in modes
(11)$$\Phi \left(x,t\right)=\sum_{k=1}^{\infty }\left[a_{k}\psi_{k}\left(x,t\right)+a_{k}^{†}\psi_{k}^{*}\left(x,t\right)\right],$$
such that the modes are given by [34]
(12)$$\psi_{k}\left(x,t\right)=\frac{i}{\sqrt{4\pi k}}\left[e^{-ik\pi G(t+x)}+e^{ik\pi F(t-x)}\right],$$
where F(z) and G(z) are functions determined by Moore’s equations
(13)$$\begin{array}{cc} G(t+L(t))-F(t-L(t)) & =0 \end{array}$$
(14)$$\begin{array}{cc} G(t+R(t))-F(t-R(t)) & =2. \end{array}$$
The functions F(z) and G(z) implement the conformal transformation
(15)$$\bar{t}+\bar{x}=G\left(t+x\right)\;\bar{t}-\bar{x}=F\left(t-x\right)$$
such that in the new coordinates, the left and right mirrors are static at ${\bar{x}}_{L}=0$ and ${\bar{x}}_{R}=1$.

Finding the evolution of the field given the motion of the mirrors is therefore reduced to solving Moore’s equations. Once this is achieved, the renormalized energy density of the field can be found [34]
(16)$$\begin{array}{c} {\langle T_{tt}\left(x,t\right)\rangle }_{\mathrm{ren}}=f_{G}\left(t+x\right)+f_{F}\left(t-x\right), \end{array}$$
where
(17)$$\begin{array}{ccc} f_{G} & = & -\frac{1}{24\pi }\left[\frac{G^{\prime \prime \prime }}{G^{\prime }}-\frac{3}{2}{\left(\frac{G^{\prime \prime }}{G^{\prime }}\right)}^{2}\right]+\frac{{\left(G^{\prime }\right)}^{2}}{2}\left[-\frac{\pi }{24}+Z\left(Td_{0}\right)\right]\\ f_{F} & = & -\frac{1}{24\pi }\left[\frac{F^{\prime \prime \prime }}{F^{\prime }}-\frac{3}{2}{\left(\frac{F^{\prime \prime }}{F^{\prime }}\right)}^{2}\right]+\frac{{\left(F^{\prime }\right)}^{2}}{2}\left[-\frac{\pi }{24}+Z\left(Td_{0}\right)\right], \end{array}$$
and $d_{0}=\left|R_{0}-L_{0}\right|$ is the initial length of the cavity. We are considering the state of the field to be initially in a thermal state at temperature *T*, and $Z(Td_{0})$ is related to the initial mean energy
(18)$$Z\left(Td_{0}\right)=\sum_{n=1}^{\infty }\frac{n\pi }{\exp\left(\frac{n\pi }{Td_{0}}\right)-1}.$$
The expression for the renormalized energy–momentum tensor above can be obtained using the standard approach based on point-splitting regularization (see for instance [35]). It can also be derived using the conformal anomaly associated with the conformal transformation Equation (Equation 15) [30,31].

Finally, it is important to note that for a static cavity with L(t)=0, $R\left(t\right)=d_{0}$, we have $F\left(z\right)=G\left(z\right)=z/d_{0}$, and the renormalized energy density reduces to the static Casimir energy density. The phenomenon of particle creation appears when F(z) and G(z) are nonlinear functions.

## 4. STA for the Field

In this case, it is particularly challenging to find an STA since the only parameters that we can control and that affect the time evolution of the field are the positions of the left and right walls, L(t) and R(t), respectively. However, we achieve this by finding the adiabatic Moore functions which correspond to the infinitely slow evolution of the field for reference trajectories $L_{\mathrm{ref}}\left(t\right)$ and $R_{\mathrm{ref}}\left(t\right)$. Then, we look for effective trajectories $L_{\mathrm{eff}}\left(t\right)$ and $R_{\mathrm{eff}}\left(t\right)$ such that they give rise to the adiabatic Moore functions previously found. The effective trajectories obtained produce an adiabatic evolution of the field in finite time, hence they constitute a shortcut to adiabaticity.

### 4.1. Adiabatic Evolution of the Field

We start by looking for functions *F* and *G* that satisfy Equations (Equation 13) and (Equation 14). We can take the derivative of the above set of equations
(19)$$G^{\prime }\left[t+L(t)\right]\left[1+\dot{L}\left(t\right)\right]-F^{\prime }\left[t-L(t)\right]\left[1-\dot{L}\left(t\right)\right]=0$$
(20)$$G^{\prime }\left[t+L(t)\right]\left[1+\dot{R}\left(t\right)\right]-F^{\prime }\left[t-R(t)\right]\left[1-\dot{R}\left(t\right)\right]=0$$
and define
(21)$$A\left(z\right):=F^{\prime }\left(z\right)$$
(22)$$B\left(z\right):=G^{\prime }\left(z\right).$$
Then, it is easy to rewrite the previous equations as
(23)$$B\left[t+L\right]\left[1+\dot{L}\right]-A\left[t-L\right]\left[1-\dot{L}\right]=0$$
(24)$$B\left[t+R\right]\left[1+\dot{R}\right]-A\left[t-R\right]\left[1-\dot{R}\right]=0.$$
Further, we can expand the functions in a Taylor series
(25)$$B\left[t+x\right]=\sum_{n}\frac{d^{n}B\left(t\right)}{dt^{n}}\frac{x^{n}}{n!}$$
(26)$$A\left[t+x\right]=\sum_{n}\frac{d^{n}A\left(t\right)}{dt^{n}}\frac{x^{n}}{n!}$$
which results in the following equations up to the third order
(27)$$\left[B+\frac{dB(t)}{dt}L+\frac{1}{2}\frac{d^{2}B\left(t\right)}{dt^{2}}L^{2}+\frac{1}{3!}\frac{d^{3}B\left(t\right)}{dt^{3}}L^{3}\right]\left[1+\dot{L}\right]-\left[A-\frac{dA(t)}{dt}L+\frac{1}{2}\frac{d^{2}A\left(t\right)}{dt^{2}}L^{2}-\frac{1}{3!}\frac{d^{3}A\left(t\right)}{dt^{3}}L^{3}\right]\left[1-\dot{L}\right]=0$$
(28)$$\left[B+\frac{dB(t)}{dt}R+\frac{1}{2}\frac{d^{2}B\left(t\right)}{dt^{2}}R^{2}+\frac{1}{3!}\frac{d^{3}B\left(t\right)}{dt^{3}}R^{3}\right]\left[1+\dot{R}\right]-\left[A-\frac{dA(t)}{dt}L+\frac{1}{2}\frac{d^{2}A\left(t\right)}{dt^{2}}L^{2}-\frac{1}{3!}\frac{d^{3}A\left(t\right)}{dt^{3}}L^{3}\right]\left[1-\dot{R}\right]=0.$$

These can be rewritten as
(29)$$B-A+{\left((B+A)L\right)}^{\prime }+\frac{1}{2}{\left(\left(B^{\prime }-A^{\prime }\right)L^{2}\right)}^{\prime }+\frac{1}{6}{\left(\left(B^{\prime \prime }+A^{\prime \prime }\right)L^{3}\right)}^{\prime }=0$$
(30)$$B-A+{\left((B+A)R\right)}^{\prime }+\frac{1}{2}{\left(\left(B^{\prime }-A^{\prime }\right)R^{2}\right)}^{\prime }+\frac{1}{6}{\left(\left(B^{\prime \prime }+A^{\prime \prime }\right)R^{3}\right)}^{\prime }=0,$$
by discarding the terms $B^{\prime \prime \prime }L^{\prime }$, $A^{\prime \prime \prime }L^{\prime }$, $B^{\prime \prime \prime }R^{\prime }$ and $A^{\prime \prime \prime }R^{\prime }$ because they involve third derivatives of time. At this point, we can expand these functions in different timescales
(31)$$A=A_{0}+A_{1}+A_{2}+A_{3}+\ldots$$
(32)$$B=B_{0}+B_{1}+B_{2}+B{}_{3}+\ldots$$
where the subindices indicate how many temporal derivatives are involved in each contribution. Using this expansion, the previous equation results for order 0 in
(33)$$A_{0}=B_{0}$$
For the first order, we have
(34)$$B_{1}-A_{1}+{\left(2A_{0}L\right)}^{\prime }=0$$
(35)$$B_{1}-A_{1}+{\left(2A_{0}R\right)}^{\prime }=0$$
and therefore
(36)$${\left(2A_{0}\left(R-L\right)\right)}^{\prime }=0\;\Rightarrow A_{0}=\frac{1}{R-L}.$$
With this result, $B_{1}-A_{1}$ can be calculated by replacing it in the previous equations.

The second order gives
(37)$$B_{2}-A_{2}+{\left(\left(A_{1}+B_{1}\right)L\right)}^{\prime }=0$$
(38)$$B_{2}-A_{2}+{\left(\left(A_{1}+B_{1}\right)R\right)}^{\prime }=0,$$
where we have used that $A_{0}=B_{0}$. Subtracting, we obtain
(39)$${\left[\left(A_{1}+B_{1}\right)\left(L-R\right)\right]}^{\prime }=0\Rightarrow A_{1}+B_{1}=\frac{k}{L-R},$$
where *k* is some constant. However, we must note that, by definition, $A_{1}$ and $B_{1}$ should have one and only one time derivative. Therefore
(40)$$k=0\Rightarrow A_{1}=-B_{1}.$$
Replacing this result in the equation for $B_{1}-A_{1}$, we find that
(41)$$A_{1}=-B_{1}={\left(A_{0}R\right)}^{\prime }={\left(\frac{R}{R-L}\right)}^{\prime }={\left(\frac{1}{2}\frac{R+L}{R-L}\right)}^{\prime }.$$
The Moore functions are then given by
(42)$$F\left(t\right)=\int dtA\left(t\right)=\int dtA_{0}\left(t\right)+\int dtA_{1}\left(t\right)+\int dtA_{2}\left(t\right)+....$$
(43)$$G\left(t\right)=\int dtB\left(t\right)=\int dtB_{0}\left(t\right)+\int dtB_{1}\left(t\right)+\int dtB_{2}\left(t\right)+....,$$
where $A_{j}\left(t\right)$ and $B_{j}\left(t\right)$ include *j* time derivatives. If the timescale in which the mirror moves is given by τ then $\int dtA_{j}\left(t\right)\propto \tau^{j-1}$, and in the adiabatic limit (τ→∞), only the first two terms are relevant. Therefore, the adiabatic Moore functions for a cavity with two moving mirrors are given by
(44)$$\begin{array}{cc} F_{\mathrm{ad}}\left(t\right) & =\int dt\frac{1}{R(t)-L(t)}+\frac{1}{2}\frac{R(t)+L(t)}{R(t)-L(t)} \end{array}$$
(45)$$\begin{array}{cc} G_{\mathrm{ad}}\left(t\right) & =\int dt\frac{1}{R(t)-L(t)}-\frac{1}{2}\frac{R(t)+L(t)}{R(t)-L(t)}. \end{array}$$

Following this procedure, one can also compute the higher adiabatic orders, generalizing to the case of two mirrors the results in Ref. [25]. However, the above results are enough for our purposes.

### 4.2. Shortcut to Adiabaticity

Given the reference trajectories for the right, $R_{\mathrm{ref}}\left(t\right)$, and left, $L_{\mathrm{ref}}\left(t\right)$, mirrors, it is possible to find effective trajectories, $R_{\mathrm{eff}}\left(t\right)$ and $L_{\mathrm{eff}}\left(t\right)$, such that the evolution of the field from start to finish is exactly the adiabatic evolution for the reference trajectories.

A way to find such effective trajectories is to select them such that the Moore functions for the field are those of the adiabatic evolution produced by the reference ones, that is
(46)$$G_{\mathrm{ad}}\left(t+L_{\mathrm{eff}}\left(t\right)\right)-F_{\mathrm{ad}}\left(t-L_{\mathrm{eff}}\left(t\right)\right)=0$$
(47)$$G_{\mathrm{ad}}\left(t+R_{\mathrm{eff}}\left(t\right)\right)-F_{\mathrm{ad}}\left(t-R_{\mathrm{eff}}\left(t\right)\right)=2,$$
where $G_{\mathrm{ad}}\left(t\right)$ and $F_{\mathrm{ad}}\left(t\right)$ are given by Equation (Equation 44) with $L\left(t\right)=L_{\mathrm{ref}}\left(t\right)$ and $R\left(t\right)=R_{\mathrm{ref}}\left(t\right)$. Thus, knowing the reference trajectories, we can solve Equations (Equation 46) and (Equation 47) independently to find effective trajectories that evolve the field in a way that exactly matches the adiabatic evolution for the reference trajectories.

### 4.3. Limit of Effective Trajectories

We would like to obtain some analytical understanding of the effective trajectories that produce the STAs. In order to do this, we exactly solve the Moore equations for the case where the reference trajectories are given by an instantaneous motion
(48)$$R_{\mathrm{ref}}\left(t\right)=R_{0}\theta \left(-t\right)+R_{f}\theta \left(t\right)$$
(49)$$L_{\mathrm{ref}}\left(t\right)=L_{0}\theta \left(-t\right)+L_{f}\theta \left(t\right).$$

We are looking for the limit effective trajectories $L_{\lim}\left(t\right)$ and $R_{\lim}\left(t\right)$ such that
(50)$$G_{\mathrm{ad}}\left(t+L_{\lim}\left(t\right)\right)-F_{\mathrm{ad}}\left(t-L_{\lim}\left(t\right)\right)=0$$
(51)$$G_{\mathrm{ad}}\left(t+R_{\lim}\left(t\right)\right)-F_{\mathrm{ad}}\left(t-R_{\lim}\left(t\right)\right)=2.$$

For these reference trajectories the adiabatic Moore functions are given by
(52)$$\begin{array}{cc} F_{\mathrm{ad}}\left(t\right) & =\frac{t+L_{0}}{R_{0}-L_{0}}\theta \left(-t\right)+\frac{t+L_{f}}{R_{f}-L_{f}}\theta \left(t\right) \end{array}$$
(53)$$\begin{array}{cc} G_{\mathrm{ad}}\left(t\right) & =\frac{t-L_{0}}{R_{0}-L_{0}}\theta \left(-t\right)+\frac{t-L_{f}}{R_{f}-L_{f}}\theta \left(t\right), \end{array}$$
which means that the Moore functions are linear functions before and after t=0. We can use this result to analyze the Moore equations one by one. If $t<-R_{0}$, then
(54)$$t-R_{\mathrm{eff}}\left(t\right)<0,\;\;t+R_{\mathrm{eff}}\left(t\right)<0$$
and the solution is
(55)$$R_{\lim}\left(t<-R_{0}\right)=R_{0}.$$
In addition, if $t>R_{f}$, then
(56)$$t-R_{\lim}\left(t\right)>0,\;\;t+R_{\lim}\left(t\right)>0$$
and the solution is
(57)$$R_{\lim}\left(t>R_{f}\right)=R_{f}.$$
However, if $-R_{0}<t<R_{f}$, then
(58)t+R(t)>0,t−R(t)<0
and the Moore equation is given by
(59)$$\frac{\left(t+R_{\lim}\left(t\right)\right)-L_{f}}{R_{f}-L_{f}}-\frac{\left(t-R_{\lim}\left(t\right)\right)+L_{0}}{R_{0}-L_{0}}=2.$$
Solving this equation for $R_{\lim}\left(t\right)$, we get
(60)$$\begin{array}{ccc} R_{\lim}\left(t\right) & = & \frac{2\left(R_{0}-L_{0}\right)\left(R_{f}-L_{f}\right)+L_{f}\left(R_{0}-L_{0}\right)+L_{0}\left(R_{f}-L_{f}\right)}{\left(R_{0}-L_{0}\right)+\left(R_{f}-L_{f}\right)}\\ & - & t\frac{\left(R_{0}-L_{0}\right)-\left(R_{f}-L_{f}\right)}{\left(R_{0}-L_{0}\right)+\left(R_{f}-L_{f}\right)}=R_{c}+v_{\lim}t. \end{array}$$
Similarly, we have $L_{\lim}\left(t<-L_{0}\right)=L_{0}$, $L_{\lim}\left(t>L_{f}\right)=L_{f}$ and
(61)$$\begin{array}{c} L_{\lim}\left(-L_{0}<t<L_{F}\right)=\frac{L_{f}\left(R_{0}-L_{0}\right)+L_{0}\left(R_{f}-L_{f}\right)}{\left(R_{0}-L_{0}\right)+\left(R_{f}-L_{f}\right)}+v_{\lim}t. \end{array}$$
In simple words, the limit effective trajectories, at early and late times, coincide with the constant position from the reference trajectory. For intermediate time values, say between these initial and final positions, the motion of the limit trajectories is simply a uniform motion with the same velocity, $v_{\lim}$, for the left and right mirrors. This velocity is determined only by the initial and final lengths of the cavity, being negative for a contraction, positive for an expansion and zero if the cavity moves rigidly.

In addition, it is possible to notice that, in general, these trajectories are not continuous functions. This is related to the fact that if the reference motion occurs in a timescale τ, there exists a critical $\tau_{c}$ (which depends on the precise reference motion) for which the effective trajectories cease to be physically achievable since the speed should be greater than the speed of light at some time.

However, by enforcing continuity for the functions,
(62)$$\begin{array}{c} R_{0}=R_{\lim}\left(-R_{0}\right),\;R_{f}=R_{\lim}\left(R_{f}\right), \end{array}$$
(63)$$\begin{array}{c} L_{0}=L_{\lim}\left(-L_{0}\right),\;L_{f}=L_{\lim}\left(L_{f}\right), \end{array}$$
we find that if $L_{f}R_{0}=L_{0}R_{f}$, the limit trajectories are actually continuous. Two simple cases where this is verified is either when there is a trivial reference motion (that is $L_{0}=L_{f}$ and $R_{0}=R_{f}$) or in the case when one of the walls is at rest at the origin, $L_{0}=L_{f}=0$, and the other moves freely.

## 5. Numerical Analysis of the STA

We now consider a particular set of reference trajectories for which we find the associated effective trajectories by numerically solving Equations (Equation 46) and (Equation 47). We consider different types of motions for the mirrors, such as a contraction, an expansion and a rigid translation, and we compare the results of the obtained trajectories and energies between the reference and effective trajectories.

Before proceeding, we need to establish a magnitude to decide whether an STA has been achieved and measure how far we are from one. Hence, we define the adiabaticity coefficient
(64)$$Q\left(t\right):=\frac{E(t)}{E_{\mathrm{ad}}\left(t\right)},$$
where E(t) is the total energy in the cavity
(65)$$\begin{array}{c} E\left(t\right)=\int_{L(t)}^{R(t)}dx{\langle T_{00}\left(x,t\right)\rangle }_{\mathrm{ren}}, \end{array}$$
while the adiabatic energy is given by
(66)$$\begin{array}{c} E_{\mathrm{ad}}\left(t\right)=-\frac{\pi }{24d}+\frac{Z(TL_{0})}{d}, \end{array}$$
where d=|R(t)−L(t)| is the length of the cavity. Notice that the adiabaticity parameter equals one if the field evolves in an adiabatic manner. However, due to the static Casimir energy (the first term of $E_{\mathrm{ad}}$), *Q* can either be lower than one for low temperatures or bigger than one for high temperatures, if the cavity is static.

Once the effective trajectories are obtained, the Moore functions are given by Equation (Equation 44). The energy and adiabaticity coefficients can then be calculated using Equations (Equation 16) and (Equation 65). However, it is useful to contrast these results with the energy and adiabaticity parameters corresponding to the original reference trajectories. In order to do this, we need to obtain the functions F(t) and G(t) by numerically solving Moore’s Equations (Equation 13) and (Equation 14). We dedicate the next section to develop an algorithm for solving this system of coupled functional equations.

### 5.1. Algorithm for Moore’s Equations

In the following, we derive an algorithm for solving Moore’s equations for F(z) and G(z) which can be used for arbitrary trajectories L(t) and R(t) of the mirrors. This algorithm is a generalization of the one used for a single moving mirror in Ref. [36].

In order to find $G(z_{1})$, we look for $t_{1}$ such that
(67)$$z_{1}=t_{1}+R\left(t_{1}\right),$$
which can be done simply by solving an algebraic equation. Then, from Equation (Equation 14) we know that
(68)$$G\left(t_{1}+R\left(t_{1}\right)\right)=F\left(t_{1}-R\left(t_{1}\right)\right)+2$$
and, solving for $t_{1}^{*}$, such that
(69)$$t_{1}-R\left(t_{1}\right)=t_{1}^{*}-L\left(t_{1}^{*}\right),$$
we find, using Equation (Equation 13),
(70)$$F\left(t_{1}^{*}-L\left(t_{1}^{*}\right)\right)=G\left(t_{1}^{*}+L\left(t_{1}^{*}\right)\right)$$
(71)$$\Rightarrow G\left(t_{1}+R\left(t_{1}\right)\right)=F\left(t_{1}^{*}-L\left(t_{1}^{*}\right)\right)+2=G\left(t_{1}^{*}+L\left(t_{1}^{*}\right)\right)+2.$$
Hence, given
(72)$$z_{2}:=t_{1}^{*}+L\left(t_{1}^{*}\right),$$
we have
(73)$$G\left(z_{1}\right)=G\left(z_{2}\right)+2.$$
If we assume $z_{2}$ to be our starting point and iterating this *n* times, we obtain
(74)$$G\left(z_{1}\right)=G\left(z_{n+1}\right)+2n.$$
Note that if L(t)<R(t) for all *t*, then
(75)$$t_{1}-R\left(t_{1}\right)=t_{1}^{*}-L\left(t_{1}^{*}\right)\Rightarrow t_{1}-t_{1}^{*}=R\left(t_{1}\right)-L\left(t_{1}^{*}\right)>0\Rightarrow t_{1}>t_{1}^{*}$$
(76)$$t_{1}^{*}+L\left(t_{1}^{*}\right)=t_{2}+R\left(t_{2}\right)\Rightarrow t_{1}^{*}-t_{2}=R\left(t_{2}\right)-L\left(t_{1}^{*}\right)>\Rightarrow t_{1}>t_{1}^{*}>t_{2},$$
which in turn means that $z_{1}>z_{2}>\ldots >z_{n}$. This means we have reduced the original problem of finding the value of the function for a given time to knowing it at a previous temporal value. We can iterate this procedure going back in time until $G(z_{n})$ is known, which eventually happens since we know the solution for static mirrors (Equation (Equation 53)). Analogously, for the other Moore function, if we wish to find $F(w_{1})$, we can search for a $t_{1}$ such that
(77)$$w_{1}=t_{1}-L\left(t_{1}\right).$$
By means of Moore’s Equation (Equation 13), we know that
(78)$$F\left(t_{1}-L\left(t_{1}\right)\right)=G\left(t_{1}+L\left(t_{1}\right)\right).$$
If we solve for ${\tilde{t}}_{1}$
(79)$${\tilde{t}}_{1}+R\left({\tilde{t}}_{1}\right)=t_{1}+L\left(t_{1}\right),$$
we obtain
(80)$$G\left({\tilde{t}}_{1}+R\left({\tilde{t}}_{1}\right)\right)=2+F\left({\tilde{t}}_{1}-R\left({\tilde{t}}_{1}\right)\right)$$
(81)$$\Rightarrow F\left(t_{1}-L\left(t_{1}\right)\right)=G\left(t_{1}+L\left(t_{1}\right)\right)=G\left({\tilde{t}}_{1}+R\left({\tilde{t}}_{1}\right)\right)=2+F\left({\tilde{t}}_{1}-R\left({\tilde{t}}_{1}\right)\right)$$
(82)$$F\left(w_{1}\right)=2+F\left({\tilde{t}}_{1}-R\left({\tilde{t}}_{1}\right)\right).$$
Finally, defining
(83)$$w_{2}:={\tilde{t}}_{1}-R\left({\tilde{t}}_{1}\right)$$
we can express the value of the function at point $w_{1}$ in terms of the value of the function at $w_{2}$
(84)$$F\left(w_{1}\right)=2+F\left(w_{2}\right).$$
In general, by iterating, we get
(85)$$F\left(w_{1}\right)=2n+F\left(w_{n+1}\right),$$
and, once more, if we go back enough times, we eventually reach the point where the mirrors are static and the function $F(z_{n})$ can be evaluated using Equation (Equation 52). In the end, we obtain an iterative algorithm to evaluate Moore’s functions F(z) and G(z) at any point.

### 5.2. Reference Trajectories

We performed a numerical analysis and compared the reference trajectories with their effective trajectories. We also computed the adiabaticity parameter for different temperatures. In order to do this, we needed a well-defined continuous energy density for the field. Since ${\langle T_{tt}\left(x,t\right)\rangle }_{\mathrm{ren}}$ involves third derivatives of the Moore functions, which in turn involve third derivatives of the reference trajectories, we chose these reference trajectories to have continuous derivatives up to the third order. We considered the motion of the wall to be restricted to a finite-time interval. In order to fulfill these conditions, we chose the reference trajectories to be
(86)$$\begin{array}{cc} L_{\mathrm{ref}}\left(t\right) & =\left(L_{f}-L_{0}\right)\delta \left(t/\tau \right)+L_{0} \end{array}$$
(87)$$\begin{array}{cc} R_{\mathrm{ref}}\left(t\right) & =R_{0}\left[1-\epsilon \delta (t/\tau )\right] \end{array}$$
where $\delta \left(x\right)=35x^{4}-84x^{5}+70x^{6}-20x^{7}$ satisfies $\delta \left(0\right)=\delta^{\prime }\left(0\right)=\delta^{\prime \prime }\left(0\right)=\delta^{\prime \prime \prime }\left(0\right)=\delta^{\prime }\left(1\right)=\delta^{\prime \prime }\left(1\right)=\delta^{\prime \prime \prime }\left(1\right)=0$ and δ(1)=1.

In the following sections, we use these trajectories to analyze different types of motions, such as a contraction, expansion and rigid motion, their effective trajectories and whether they achieve an STA.

### 5.3. Contraction

We first analyzed a symmetric contraction of the cavity, meaning that both mirrors performed the same reference motion at the same time, but in opposite directions. We represented this by considering the reference functions Equations (Equation 86) and (Equation 87) and solving numerically Equations (Equation 46) and (Equation 47) for the effective trajectories, $R_{\mathrm{eff}}\left(t\right)$ and $L_{\mathrm{eff}}\left(t\right)$.

In Figure 2a, we show the reference and corresponding effective trajectories for the left (dashed lines) and right mirrors (solid lines) in a symmetric contraction. We note that the effective trajectory for the right mirror starts moving first close to $t=-R_{0}$, while the left mirror moves at later times near $t=-L_{0}$, as pointed out by our analysis for the limit trajectories. If we look at $R_{\mathrm{eff}}$, we also note the local minimum and maximum around these points develop into discontinuities for very small τ. This can also be seen in Figure 3, where we compare the effective trajectories for an asymmetric contraction with $\tau /R_{0}=0.4$ and the limit effective trajectories analytically (Equations (Equation 60) and (Equation 61)). Therein, the discontinuities are more evident. It is also noticeable that the right trajectory converges faster than the left one and that the slope of the curve, i.e., the velocity, is negative, which is consistent with our analytical results.

In Figure 2b, we show the resulting Moore functions for the reference and effective trajectories. In the case of the effective trajectory, the functions are linear at the early and late times. On the other hand, Moore’s functions for the reference trajectory are linear plus an oscillation at late times, which is the manifestation of particle creation.

Further, we analyzed the adiabaticity parameter for different initial temperatures as shown in Figure 4. We note that the adiabaticity parameter is initially one. However, for both the reference and effective trajectories at later times, the reference trajectory is very far from unity, while the effective trajectory returns to one, indicating that an adiabatic evolution has been achieved by reabsorbing the emitted photons. It is also noticeable that as the temperature increases, the curves become smoother.

### 5.4. Expansion

We then analyzed our proposed STA for a reference trajectory given by a symmetric expansion of the cavity. To achieve this, we used the reference trajectories given by Equations (Equation 86) and (Equation 87) with ϵ<0 and $L_{f}=\epsilon R_{0}$.

In Figure 5a, we show the reference and corresponding effective trajectories for the left (dashed lines) and right (solid lines) mirrors in a symmetric expansion. We notice that the effective trajectory of the right mirror has a local minimum and maximum close to the point where a discontinuity will develop for τ→0, which is in agreement with the limit effective trajectories calculated. On the other hand, the effective trajectory of the right mirror is very similar to the reference one. This is because, as we have previously seen, the convergence of the effective trajectory of the left mirror to the limit is slower. The Moore functions, however, have a similar behavior to that of the previous case.

In Figure 6, we present the adiabaticity parameter for a symmetric expansion for three different temperatures. In this case, the adiabaticity parameter again confirms that we have obtained an STA, as it is equal to one for late times for the effective trajectories. We can also see that the effect of the temperature on this parameter is to smooth the curve as the temperature increases. The STA allows us to save more energy for higher temperatures.

### 5.5. Rigid Motion

The final type of reference trajectory that we considered was a rigid translation. To achieve this, we used the reference trajectories given by Equations (Equation 86) and (Equation 87) with ϵ<0 and $L_{f}=-\epsilon R_{0}$. There were several motivations for this from the fact that the limit effective trajectories were qualitatively different from fundamental questions on relativistic quantum information tasks.

In Figure 7, we show the reference and corresponding effective trajectories for the left (dashed lines) and right (solid lines) mirrors in a rigid translation. We can see that the effective trajectory of the left mirror is very similar to the reference trajectory. However, the effective trajectory for the right mirror is extremely different from the other two cases studied previously. We observe that the right mirror moves half of the way while the left one is static, then it stops, and the left mirror moves and stops, and then it moves again up to the final position. Although this motion can look strange at first sight, it is very well described by the limit effective trajectory in Figure 8, which predicts that the speed of the motion should be zero for a reference trajectory that does not change the length of the cavity. The Moore functions also have a similar behavior as in the previous cases.

Finally, we studied the adiabaticity parameter for the reference rigid motion as shown in Figure 9. We see that the effective trajectories given by Equations (Equation 86) and (Equation 87) result in a successful shortcut to adiabaticity, since for late times, $Q_{\mathrm{eff}}=1$. The energy saved by using this motion protocol is greatly enhanced for high temperatures for the initial state of the quantum field.

## 6. Discussion

As technology improves and quantum systems can be operated at smaller timescales, it becomes increasingly important to consider the nonadiabatic effects of these operations and to develop new ways of mitigating of even avoiding them entirely. With this motivation in mind, in this manuscript we found a shortcut to adiabaticity for a scalar quantum field in a one-dimensional cavity with two moving mirrors. This allowed an extremely efficient way to manipulate microwave resonators very rapidly. Moreover, our results gave an explicit protocol to find STAs for any initial and final state of the mirrors, which can be implemented in an experimental setup by choosing adequately effective trajectories for the mirrors calculated from a given reference trajectory.

We analytically analyzed the properties of these effective trajectories and found that the limit effective trajectories, for infinitely fast reference trajectories, are in general not continuous functions, which signaled that there was a critical timescale beyond which the resulting shortcut ceased to be physical, since the mirror should not move faster than the speed of light.

In addition, we solved numerically the effective trajectories for three different types of reference motions: a contraction, an expansion and a rigid translation of the cavity. Our numerical analysis confirmed the analytical results, showing that our protocol successfully implemented a shortcut to adiabaticity and that the effective trajectories were very well described by the limit effective trajectories found analytically.

These findings call for further studies that analyze in more depth the experimental implementation in superconducting circuits or optomechanical cavities. It would also be interesting to better understand the instantaneous energetic cost of this STA and their utilization in more efficient quantum heat engines.

## Figures and Tables

**Figure 1 entropy-25-00018-f001:**
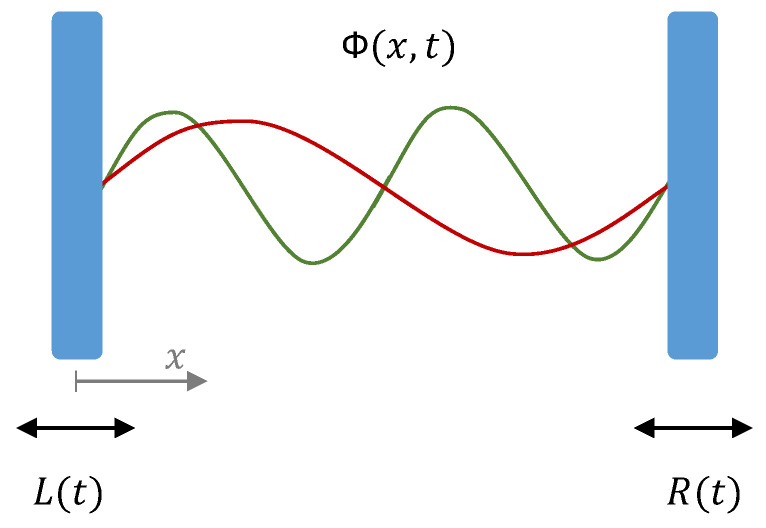
Schematics of the one dimensional cavity with a scalar quantum field Φ(x,t) inside and two moving mirrors with trajectories L(t) and R(t). The red and green curves illustrate two modes of the field in the cavity.

**Figure 2 entropy-25-00018-f002:**
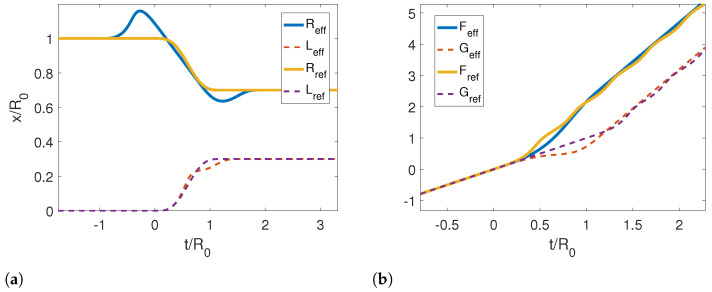
(**a**) Reference and corresponding effective trajectories for the left and right mirrors in the case of a symmetric contraction. (**b**) Resulting Moore’s functions for reference and effective trajectories. The parameters used for this calculation were $\tau /R_{0}=1.2$, ϵ=0.3, $L_{0}/R_{0}=0$, $L_{f}/R_{0}=0.3$ and $R_{f}/R_{0}=0.7$.

**Figure 3 entropy-25-00018-f003:**
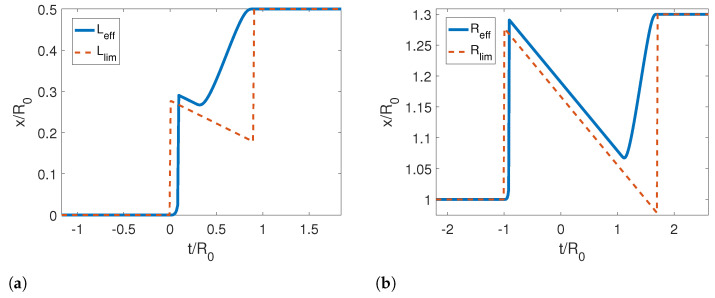
(**a**) Effective and limit trajectories for the left mirror for an asymmetric contraction. (**b**) Effective and limit effective trajectories for the right mirror. The parameters used for this calculation were $\tau /R_{0}=0.4$, ϵ=−0.3, $L_{0}/R_{0}=0$, $L_{f}/R_{0}=0.5$ and $R_{f}/R_{0}=1.3$.

**Figure 4 entropy-25-00018-f004:**
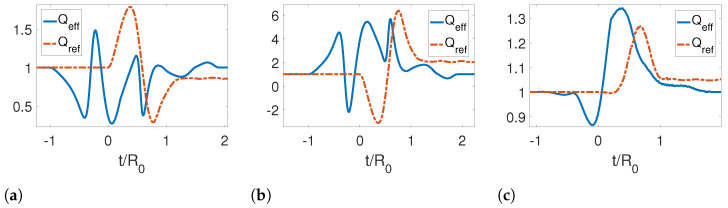
Adiabaticity parameter for a symmetric contraction for three different temperatures: (**a**) $TR_{0}=0$, (**b**) $TR_{0}=1$ and (**c**) $TR_{0}=5$. The parameters used for this calculation were $\tau /R_{0}=1.2$, ϵ=0.3, $L_{0}/R_{0}=0$ and $L_{f}/R_{0}=0.3$, $R_{f}/R_{0}=0.7$.

**Figure 5 entropy-25-00018-f005:**
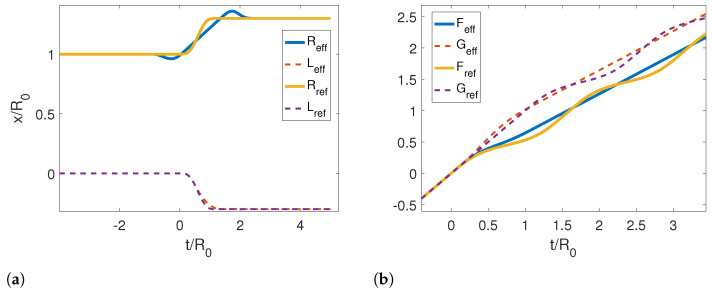
(**a**) Reference and corresponding effective trajectories for the left and right mirrors in the case of a symmetric expansion. (**b**) Resulting Moore functions for reference and effective trajectories. The parameters used for this calculation were $\tau /R_{0}=1.2$, ϵ=−0.3, $L_{0}/R_{0}=0$, $L_{f}/R_{0}=-0.3$ and $R_{f}/R_{0}=1.3$.

**Figure 6 entropy-25-00018-f006:**
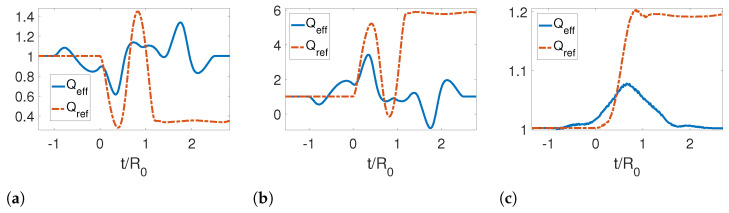
Adiabaticity parameter for a symmetric expansion for three different temperatures: (**a**) $TR_{0}=0$, (**b**) $TR_{0}=1$ and (**c**) $TR_{0}=5$. The parameters used for this calculation were $\tau /R_{0}=1.2$, ϵ=−0.3, $L_{0}/R_{0}=0$, $L_{f}/R_{0}=-0.3$ and $R_{f}/R_{0}=1.3$.

**Figure 7 entropy-25-00018-f007:**
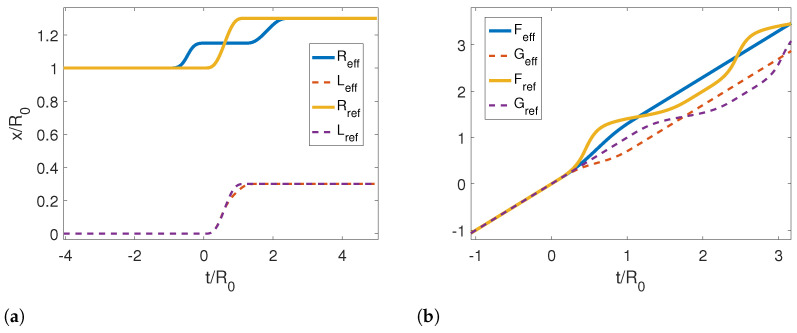
(**a**) Reference and corresponding effective trajectories for the left and right mirrors in the case of a rigid translation. (**b**) Resulting Moore functions for reference and effective trajectories. The parameters used for this calculation were $\tau /R_{0}=1.2$, ϵ=−0.3, $L_{0}/R_{0}=0$ and $L_{f}/R_{0}=0.3$$R_{f}/R_{0}=1.3$.

**Figure 8 entropy-25-00018-f008:**
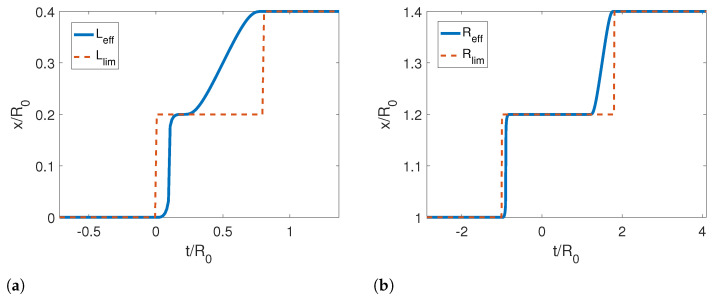
(**a**) Effective and limit trajectories for the left mirror for a rigid motion. (**b**) Effective and limit trajectories for the right mirror for a rigid motion. The parameters used for this calculation were $\tau /R_{0}=0.4$, ϵ=−0.4, $L_{0}/R_{0}=0$, $L_{f}/R_{0}=0.4$ and $R_{f}/R_{0}=1.3$.

**Figure 9 entropy-25-00018-f009:**
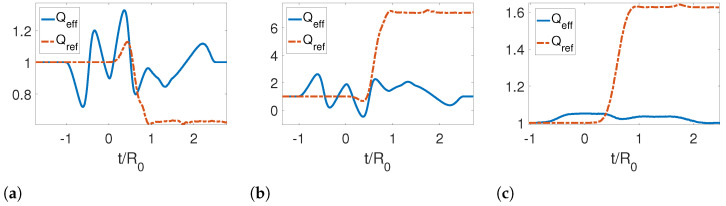
Adiabaticity parameter for a rigid translation for three different temperatures: (**a**) $TR_{0}=0$, (**b**) $TR_{0}=1$ and (**c**) $TR_{0}=5$. The parameters used for this calculation were $\tau /R_{0}=1.2$, ϵ=−0.3, $L_{0}/R_{0}=0$, $L_{f}/R_{0}=0.3$ and $R_{f}/R_{0}=1.3$.

## Data Availability

Not applicable.

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
