# Peer review of "Fast Adiabatic Control of an Optomechanical Cavity"

_entropy, 2022, doi:10.3390/e25010018_

Round 1

Reviewer 1 Report

The manuscript by Del Grosso and coworkers, entitled 'Fast adiabatic control of an optomechanical cavity', derives control functions to manipulate the quantum fields inside a cavity using shortcut to adiabaticity protocols (STA). They do so in the framework of two moving mirrors forming the optomechanical cavities, allowing for two control fields. This derivation is a continuation of their earlier paper of February of this year published in PRA. The authors first introduce general concepts of STA, before moving on to the specific case of STA on a quantum fields and them derive and test protocols for STA on a two moving mirrors optomechanical cavity.

The paper addresses a question that is for now theorical, 'can one control adiabatically a quantum field in a finite time?', which in the case of an optomechanical cavity could be reformulated as whether one can control multimode states adiabatically. This has to my knowledge not been addressed theoretically or experimentally in the terms set by the paper, and is an interesting questions in the perspectives of quantum control over increasingly complex quantum systems, and probing quantum field theory using quantum systems.

The manuscript is clearly written and layout. The acknowledgement and difference compared to the previous work is clearly made, and to me represent a small enough overlap to not diminish the suitability of the present paper for publication. Yet, I wonder if the authors could comment whether having two controllable mirrors versus one improves, fasten, hinder or just complexify the STA protocol.

The explanation of the derivations of the control functions is well-done, with hypothesis clearly written. I have some questions concerning the numerical analysis as well as the critical time-scale. From the reference functions given as initial guess, the effective funtions vary in effective duration by a factor 1 to 3: what set this time-scale ? What is the limit time-wise for obtaining adiabacity ?

The final functions show many discontinuity - how stable is the solution, for example if one would continue at times t>>1/R0 ? Would enforcing second-order derivative continuity be possible and still yield physical solutions ?

Other comments:
Eq 4-5: some variables are not defined
I would remove superconducting resonator as one of the keywords of the paper, since nothing in the text except vague references relates to superconducting resonators.

Reviewer 2 Report

see attached pdf file

Author Response

please find the attache file

Round 2

Reviewer 2 Report

The authors have introduced the minor modifications requested and have satisfactorily replied to my queries. I find the paper now suitable for publication.